# Estrogen regulates sex-specific localization of regulatory T cells in adipose tissue of obese female mice

**Akari Ishikawa**[1]☯, **Tsutomu Wada**[1]☯*, **Sanshiro Nishimura**[1], **Tetsuo Ito**[1], **Akira Okekawa**[1], **Yasuhiro Onogi**[1], **Eri Watanabe**[1], **Azusa Sameshima**[2], **Tomoko Tanaka**[2], **Hiroshi Tsuneki**[1], **Shigeru Saito**[2], **Toshiyasu Sasaoka**[1]*

**1** Department of Clinical Pharmacology, University of Toyama, Toyama, Japan, **2** Department of Obstetrics and Gynecology, University of Toyama, Toyama, Japan

☯ These authors contributed equally to this work.
* twada@pha.u-toyama.ac.jp (TW); tsasaoka@pha.u-toyama.ac.jp (TS)

**Data Availability Statement:** All relevant data are within the manuscript and its Supporting Information file.

## Abstract

Regulatory T cells (Treg) play essential roles in maintaining immune homeostasis. Resident Treg in visceral adipose tissue (VAT-Treg) decrease in male obese mice, which leads to the development of obesity-associated chronic inflammations and insulin resistance. Although gender differences in immune responses have been reported, the effects of the difference in metabolic environment on VAT-Treg are unclear. We investigated the localization of VAT-Treg in female mice in comparison with that in male mice. On a high-fat diet (HFD), VAT-Treg decreased in male mice but increased in female mice. The increase was abolished in ovariectomized and HFD-fed mice, but was restored by estrogen supplementation. The IL33 receptor ST2, which is important for the localization and maturation of VAT-Treg in males, was reduced in CD4$^+$CD25$^+$ T cells isolated from gonadal fat of obese mice of both genders, suggesting that a different system exists for VAT-Treg localization in females. Extensive analysis of chemokine expression in gonadal fat and adipose CD4$^+$CD25$^+$T cells revealed several chemokine signals related to female-specific VAT-Treg accumulation such as CCL24, CCR6, and CXCR3. Taken together, the current study demonstrated sexual dimorphism in VAT-Treg localization in obese mice. Estrogen may attenuate obesity-associated chronic inflammation partly through altering chemokine-related VAT-Treg localization in females.

## Introduction

Numerous biological phenomena exhibit distinctive traits due to sexual dimorphism. Immune and metabolic systems are representative because they are markedly affected by sex hormones [1, 2]. As systemic metabolism is regulated by immune systems, their coordinated association has been explored in immuno-metabolism studies. Obesity-associated infiltration of immune cells promotes chronic inflammation, especially in visceral adipose tissue, and exacerbates insulin resistance and glucose metabolism [3, 4]. However, our understanding of the sexual

**Funding:** This study was funded by the Japan Society for the Promotion of Science (JSPS KAKENHI Grant Number JP15K09410) to TW and a research grant from Mitsubishi Tanabe Pharma Corporation to TW. The funders had no role in study design, data collection and analysis, decision to publish, or preparation of the manuscript.

**Competing interests:** This study was funded in part by a research grant from Mitsubishi Tanabe Pharma Corporation to TW. This does not alter our adherence to PLOS ONE policies on sharing data and materials.

dimorphism in immune cells is currently limited regarding the development of obesity-associated chronic inflammation.

Regulatory T cells (Treg) are a specific subpopulation of T cells that can suppress inappropriate or extreme immune responses such as autoimmune reactions [5]. Increased estrogen plays a significant role in establishing maternal-fetal immune tolerance during pregnancy by promoting the differentiation of Treg from naïve T cells [6]. In contrast, the number of Treg decreases in the visceral adipose tissue of male obese mice [7, 8]. As Treg can alleviate obesity-associated chronic inflammation, their reduction further exacerbates chronic inflammation in obesity. Indeed, increased adipose inflammation was observed in Treg-depleted mice, whereas glucose metabolism was ameliorated in obese mice after adopted transfer of Treg [7, 9]. Although several underlying mechanisms of reduced adipose Treg in obesity have been examined, they were investigated only in males. Males are more prone to metabolic abnormalities than females upon overnutrition due to the absence of estrogen, a cardinal female hormone [10]. However, the impact of obesity on adipose Treg in females and the effects of estrogen on their regulation are unknown.

Tissue resident Treg exhibit distinct gene expression profiles that markedly affect their immune properties. Visceral adipose tissue-localized Treg (VAT-Treg) are one of most well-characterized tissue-resident Treg subsets that distinctively express peroxisome proliferator activated receptor γ (PPARγ) and related genes [11]. In addition, VAT-Treg have been demonstrated to have unique chemokine expression profiles [7, 11], suggesting the existence of unknown chemokine signals related to VAT-Treg accumulation. Indeed, tissue-specific chemokine signals have been identified for the regulation of tissue Treg distribution in several tissues [12–14]. However, the regulatory mechanism of Treg trafficking in adipose tissue during obesity development is unclear, especially in females.

VAT-Treg play a significant role in adipose chronic inflammation and systemic glucose metabolism in obesity. In the current study, we aimed to clarify the relationship among glucose metabolism, chronic inflammation, and Treg distribution in the adipose tissue of lean and obese mice from the viewpoint of sex difference. We found an increase in VAT-Treg in obese female mice. Therefore, we concurrently prepared ovariectomized mice and estrogen-supplemented mice fed a high-fat diet (HFD) to examine the impact of estrogen on obese phenotypes and Treg distribution. Moreover, the expression of chemokines and their receptors suspected to function in tissue-trafficking of Treg was examined by analyzing isolated CD4$^+$CD25$^+$ T cells from VAT of both genders. Thus, the current study demonstrated the sexual dimorphism and impact of estrogen on VAT-Treg accumulation during obesity development, and the suggested chemokine signals as the underlying mechanism.

## Materials and methods

### Animals and experimental groups

Eight-week-old male and female C57BL/6J mice were purchased from Japan SLC (Shizuoka, Japan). They were divided into six experimental groups: 1) male mice fed normal diet (Rodent Diet 20 5053; LabDiet, St. Louis, MO, USA) (M-Chow), 2) male mice fed 60 kcal% high-fat diet (D12492; Research Diets, New Brunswick, NJ, USA) (M-HFD), 3) female mice fed normal diet (F-Chow), 4) female mice fed 60 kcal% HFD (F-HFD), 5) ovariectomized female mice fed 60 kcal% HFD (OVX-HFD), and 6) OVX-HFD mice receiving estradiol (OVX-HFD+E2). Ovariectomy and sham-operation were performed under anesthesia with pentobarbital sodium. Mice were housed under a 12:12-h light-dark cycle (lights on at 07:00) in a temperature-controlled colony room, and were provided food and water ad libitum. Mice were fasted overnight and euthanized by cervical dislocation for analysis. All experimental procedures

used in this study were approved by the Committee of Animal Experiments at University of Toyama (A2013PHA-15 and A2016PHA-14).

## Exogenous estradiol treatment

17β-estradiol (Sigma-Aldrich, Darmstadt, Germany) or vehicle was administered via subcutaneously injection (1.5 μg/mice in sesame oil) every 4 days by imitating the estrus cycle of mice, referring to a previous study on rats with minor modification [15]. The dose of estrogen was determined based on the body weight transition after the treatment.

## Analysis of body composition

Body fat composition was analyzed by magnetic resonance imaging (MRI) under anesthesia in mice at 12 weeks after the initiation of HFD feeding, as described previously [16]. Series of T1-weighted axial slices were analyzed using ImageJ (NIH, Bethesda, MD, USA).

## Analysis of energy metabolism

Oxygen consumption ($VO_2$), the production of carbon dioxide ($VCO_2$), and locomotor activity in mice were measured in metabolic chambers (MK-5000RQ, Muromachi Kikai, Tokyo, Japan) with free access to food and water, as described previously [16].

## Glucose and insulin tolerance test

The glucose tolerance test (GTT) and insulin tolerance test (ITT) were conducted on mice 12–13 or 15 weeks after the initiation of HFD feeding. For the GTT, mice fasted for 6 h were injected intraperitoneally with glucose (2g/kg body weight). For the ITT, mice fasted for 2 h were injected intraperitoneally with insulin (0.75 U/kg body weight) [17, 18].

## Isolation of stromal-vascular fraction

Gonadal white adipose tissues (Wg) of mice were minced and digested with collagenase (Wako Pure Chemical Industries Ltd, Osaka, Japan) at 37˚C for 60 min. Samples were passed through mesh, centrifuged at 220 xg for 15 min, and the stromal-vascular fraction (SVF) was isolated as a pellet. Pellets were rinsed twice and incubated in lysing buffer (BD Biosciences) for 15 min, and the SVF was used for flow cytometry.

## Isolation of splenocytes

Spleens were grinded with slide glasses. Samples were passed through mesh, centrifuged at 220 xg for 15 min, and pellets were incubated in lysing buffer for 1 min. Then, splenocytes were subjected to flow cytometry analysis.

## Flow cytometry analysis

SVF cells and splenocytes were incubated with purified rat anti-mouse CD16/CD32 (BD Biosciences, San Jose, CA, USA) for 15 min, and then stained with antibodies or the matching isotype controls. For the analysis of 7AAD⁻CD45⁺CD4⁺CD8⁻CD25⁺FOXP3⁺ Tregs, SVF cells and splenocytes were stained with PE-Cy7 anti-mouse CD45 antibody (eBioscience), FITC rat anti-mouse CD4 antibody (BD Biosciences), APC-Cy7 anti-mouse CD8a antibody (BioLegend), and PE anti-mouse CD25 antibody (BioLegend) for 40 min. Cells were rinsed and incubated with 7-amino-actinomycin D (BD Biosciences) for 15 min. Cells were rinsed twice and fixed in 4% paraformaldehyde (Wako Pure Chemical Industries Ltd) for 15 min. After

washing, cells were kept at 4°C overnight. The next day, cells were permeabilized with 0.1% polyoxyethylene sorbitan monolaurate (Wako Pure Chemical Industries Ltd) for 20 min. After washing, cells were incubated with purified rat anti-mouse CD16/CD32 for 15 min and stained with APC anti-mouse/rat Foxp3 (eBioscience) for 60 min. Then, the numbers of Treg were analyzed by FACSAria II (BD Biosciences). Data were analyzed by FACS Diva 6.1.2 (BD Bioscience) or FCS Express (De Novo Software). The gating strategy of Treg cells are shown in S1 Fig. For the isolation of 7AAD⁻CD45⁺CD4⁺CD8⁻CD25⁺ cells (CD4⁺CD25⁺T cells), SVF cells and splenocytes were stained with PE-Cy7 anti-mouse CD45 antibody (eBioscience), APC rat anti-mouse CD4 antibody (BD Biosciences), FITC rat anti-mouse CD8a antibody (BD Biosciences), and PE anti-mouse CD25 antibody (BioLegend). This fraction was isolated by FACSAria II and subjected to real-time PCR analysis.

## Real-time quantitative PCR

RNA extraction, reverse transcription, and real-time PCR using SYBR green were performed as previously described [17]. The relative expression of objective mRNA was calculated as a ratio to that of the 18S ribosomal RNA. Primer sequences are listed in Table 1.

## Statistical analysis

Data are expressed as the mean ± S.E. Statistical analysis was performed using the Student's *t*-test between two groups or one-way ANOVA and Bonferroni test for multiple comparisons using the software ystat2004. Statistical analysis for body weight transition, blood glucose levels

**Table 1. Primer list.**

| Genes | Forward primer | Reverse primer |
|---|---|---|
| *Emr1* | CTTTGGCTATGGGCTTCCAGTC | GCAAGGAGGACAGAGTTTATCGTG |
| *Itgax* | ATGTTGGTGGAAGCAAATGG | CCTGGGAATCCTATTGCAGA |
| *Tnfa* | AGCCTGTAGCCCACGTCGTA | GGCACCACTAGTTGGTTGTCTTTG |
| *Il1b* | TCCAGGATGAGGACATGAGCAC | GAACGTCACACACCAGCAGGTTA |
| *Il33* | CCTGCCTCCCTGAGTACATACA | CTTCTTCCCATCCACACCGT |
| *Il1rl1* | GCAATTCTGACACTTCCCATG | ACGATTTACTGCCCTCCGTA |
| *Ccl2* | TCACCTGCTGCTACTCATTCACCA | TACAGCTTCTTTGGGACACCTGCT |
| *Ccl3* | TGAAACCAGCAGCCTTTGCTC | AGGCATTCAGTTCCAGGTCAGTG |
| *Ccl5* | CCTCACCATCATCCTCACTGCA | TCTTCTCTGGGTTGGCACACAC |
| *Ccl11* | TTCTATTCCTGCTGCTCACGG | AGGGTGCATCTGTTGTTGGTG |
| *Ccl20* | CGACTGTTGCCTCTCGTACA | GAGGAGGTTCACAGCCCTTT |
| *Ccl21* | TGAGCTATGTGCAAACCCTGAGGA | TGAGGGCTGTGTCTGTTCAGTTCT |
| *Ccl22* | TCTTGCTGTGGCAATTCAGA | GAGGGTGACGGATGTAGTCC |
| *Ccl24* | CTGTGACCATCCCCTCATCT | TATGTGCCTCTGAACCCACA |
| *Cxcl10* | TGCTGGGTCTGAGTGGGACT | CCCTATGGCCCTCATTCTCAC |
| *Ccr1* | TTAGCTTCCATGCCTGCCTTATA | TCCACTGCTTCAGGCTCTTGT |
| *Ccr2* | AGAGGTCTCGGTTGGGTTGT | CACTGTCTTTGAGGCTTGTTGC |
| *Ccr3* | TTTCCTGCAGTCCTCGCTAT | ATAAGACGGATGGCCTTGTG |
| *Ccr4* | CGAAGGTATCAAGGCATTTGGG | GTACACGTCCGTCATGGACTT |
| *Ccr5* | ATACCCGATCCACAGGAGAA | CCATTCCTACTCCCAAGCTG |
| *Ccr6* | TTGTCCTCACCCTACCGTTC | GATGAACCACACTGCCACAC |
| *Ccr7* | CCAGCAAGCAGCTCAACATT | GCCGATGAAGGCATACAAGA |
| *Cxcr3* | GCCAAGCCATGTACCTTGAG | GGAGAGGTGCTGTTTTCCAG |
| *18s rRNA* | GTAACCCGTTGAACCCCATT | CCATCCAATCGGTAGTAGCG |

in GTT and ITT in female mice were conducted by two-way ANOVA and Bonferroni test for multiple comparisons using the software StatView5.0. P<0.05 was considered significant.

## Results

### Sex difference in energy and glucose metabolism of diet-induced obesity

The impact of HFD feeding differs between male and female [2]. Therefore, we initially examined the metabolic profiles of each mouse. In males, HFD feeding strongly augmented body weight gain and fat accumulations in both gonadal and inguinal white adipose tissue (Wg and Wi, respectively). These increases were correlated with a decrease in $VCO_2$ in the dark phase (Fig 1A–1E). In females, F-HFD mice exhibited a significant increase in body weight compared with those fed F-Chow, albeit to a lesser extent than in male mice. In contrast, the body weight of OVX-HFD mice significantly increased, whereas replacement of E2 effectively attenuated weight gain to a similar level as F-HFD mice (Fig 1F). Similar changes were observed in Wi weights at sacrifice, and in visceral and subcutaneous adipose tissue volumes analyzed by MRI at 12 weeks of HFD feeding; however, Wg weights at sacrifice were almost similar among the three groups of HFD-fed mice (Fig 1I–1M). Regarding energy metabolism, $VO_2$ in the dark phase, and $VCO_2$ in both light and dark phases decreased in the HFD-fed female mice. Of note, OVX-HFD mice exhibited further reduction of $VO_2$ in the dark phase compared with F-HFD mice, and E2 treatment ameliorated this reduction (Fig 1G and 1H).

We next investigated the glucose metabolism by GTT and ITT (Fig 2). M-HFD mice exhibited significantly higher blood glucose levels by both GTT and ITT, suggesting glucose intolerance and insulin resistance (Fig 2A and 2B). In contrast, F-HFD mice exhibited a modest increase in glucose levels by GTT and ITT. However, glucose levels were significantly increased, and E2 treatment attenuated this increase in OVX-HFD mice (Fig 2C and 2D).

### Sex difference in adipose Treg localization

VAT-Treg play an essential role in the regulation of chronic inflammation and glucose homeostasis, and this reduction is considered as a causative factor promoting chronic inflammation in male obese mice [7]. However, little is known about the impact of obesity on VAT-Treg in females. To elucidate the sex-specific properties of adipose-resident Treg localization, we analyzed Treg in the spleen and VAT by flow cytometry. In the spleen, CD4+ T cells in female OVX and OVX-HFD mice, and CD8+ T cells in all mice fed HFD were decreased. In contrast, the number of Treg did not differ among mouse groups of either sex (Fig 3A and 3B). On the other hand, CD4+ T cells were reduced and CD8+ T cells were increased by HFD feeding in the Wg of male mice (Fig 3C) as previously reported [8,19]. However, no such changes were observed in female mice (Fig 3D). Importantly, VAT-Treg were significantly reduced by HFD in male mice (Fig 3C) as previously reported [7, 8], whereas their number was slightly and significantly increased in F-HFD and OVX-HFD+E2 mice, and not altered in OVX-HFD mice (Fig 3D).

### Sex difference in chronic inflammation of adipose tissue

We next examined mRNA expression of genes related to chronic inflammation in the Wg. In M-HFD males, the expression of macrophage markers *Emr1* and *Itgax*, and proinflammatory cytokines *Tnfa* and *Il1b* were significantly increased (Fig 4A–4D). These findings are consistent with the reduction of VAT-Treg in M-HFD mice. However, in females, the increase in these genes was observed only in OVH-HFD mice, and not in F-HFD or OVX-HFD+E2 mice (Fig 4E–4H). Therefore, the lower expression of proinflammatory genes in these mice was

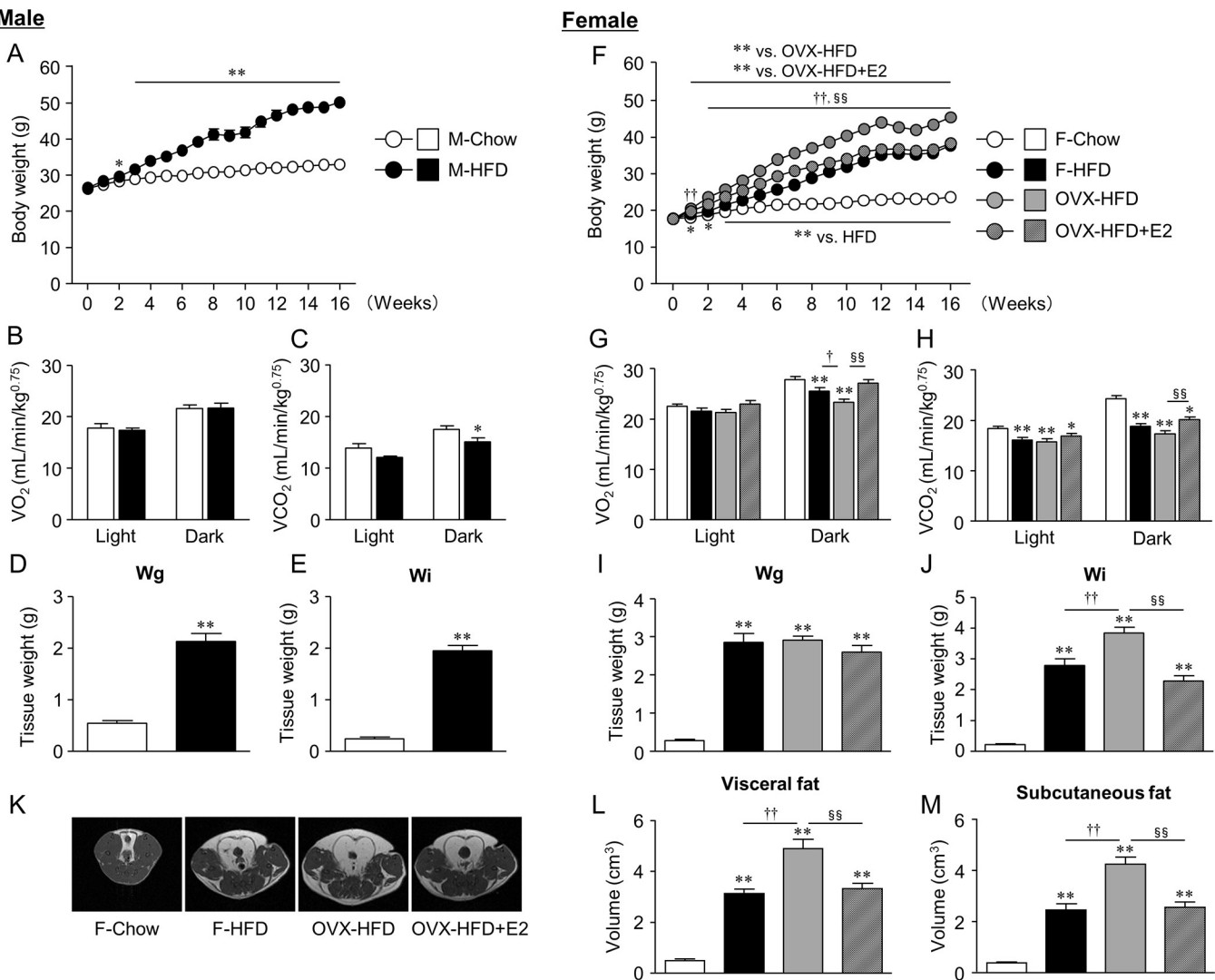

**Fig 1. Sex difference in metabolic phenotypes of diet-induced obesity.** Changes in body weight (A, F), oxygen consumption (B, G), carbon dioxide production (C, H), weights of gonadal (Wg) and inguinal white adipose tissue (Wi) (D, E, I, J) in male (A-E) and female mice (F-J) are shown. Representative T1-weighted axial MRI slices of female mice (K), and estimated volumes of visceral (L) and subcutaneous fat (M) in each experimental group of female mice are shown. Data are the mean ± S.E. ($n$ = 10–18 in A, F; $n$ = 5–9 in B-E, G-M). $^*P<0.05$ and $^{**}P<0.01$, significantly different from Chow mice; $^†P<0.05$ and $^{††}P<0.01$, significantly different between F-HFD and OVX-HFD mice; $^§P<0.05$ and $^{§§}P<0.01$, significantly different between OVX-HFD and OVX-HFD+E2 mice.

consistent with the higher accumulation of VAT-Treg in F-HFD and OVX-HFD+E2 mice (Fig 3D).

## Comprehensive expression analysis of chemokines and their receptors in the Wg and accumulated CD4$^+$CD25$^+$ T cells

Interleukin 33 (IL33) signaling through its receptor ST2 (Interleukin 1 receptor-like 1; Il1rl1) plays an important role in the recruitment and maintenance of VAT-Treg in males [20, 21]. Therefore, we analyzed the expression of *Il33* in the Wg and *Il1rl1* in adipose CD4$^+$CD25$^+$ T cells (Fig 5). Consistent with IL33 being considered as 'alarmin', its expression was higher in the Wg of M-HFD mice than in that of M-Chow mice (Fig 5A), suggesting inflammation and

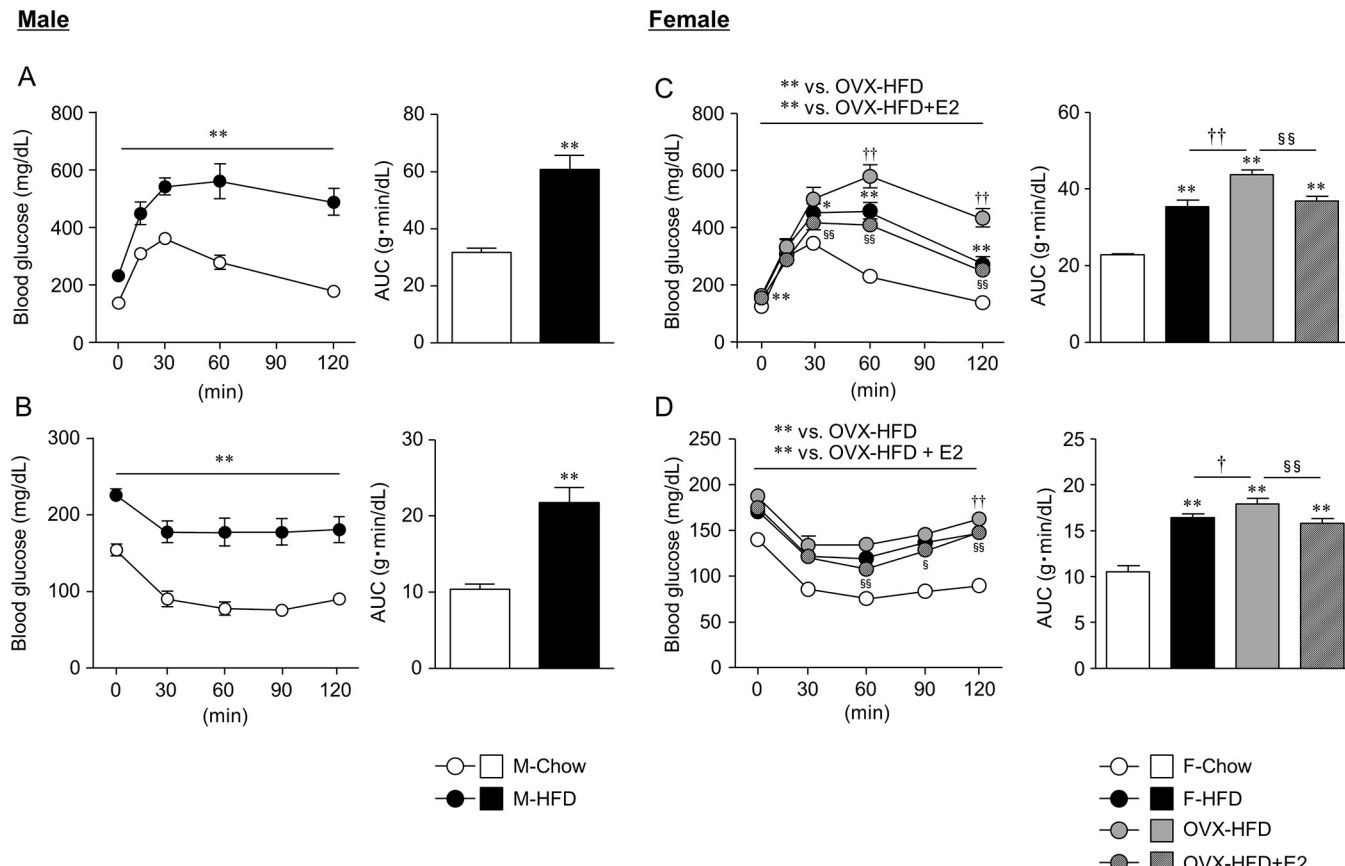

**Fig 2. Sex difference in glucose metabolism of diet-induced obesity.** Glucose tolerance test (A, C), insulin tolerance test (B, D), and glucose area under the curve (AUC) in male and female mice are shown. Data are the mean ± S.E. ($n$ = 5–9). $^*P<0.05$ and $^{**}P<0.01$, significantly different from control mice; $^{†}P<0.05$ and $^{††}P<0.01$, significantly different between F-HFD and OVX-HFD mice; $^{§}P<0.05$ and $^{§§}P<0.01$, significantly different between OVX-HFD and OVX-HFD+E2 mice.

tissue damage associated with obesity [20]. These effects were similar in females; therefore, *Il33* expression in Wg was correlated with the degree of obesity and inflammation (Fig 5C), which was significantly higher in OVX-HFD female mice among the female mouse groups (Figs 1 and 4). In contrast, the *Il1rl1* expression in adipose CD4+CD25+ T cells slightly decreased in M-HFD mice (Fig 5B), consistent with the previous observation in male obese mice [20]. Similarly, a decrease in *Il1rl1* expression in adipose CD4+CD25+ T cells was observed in both F-HFD and OVX-HFD mice (Fig 5D).

The impact of HFD feeding on VAT-Treg accumulation differed between male and female mice (Fig 3). Therefore, we hypothesized a sex-specific trafficking mechanism of VAT-Treg. As VAT-Treg express several chemokine receptors that are suggested to be involved in the migration and extravasation of tissue-resident Treg [7, 11], we examined the expression of chemokines in the Wg and their corresponding receptors in adipose CD4+CD25+ T cells (Fig 6). The heat map of chemokine expression in male mice revealed that most chemokines were increased in the Wg of M-HFD mice, whereas that of their receptors in adipose CD4+CD25+ T cells was lower than that in M-Chow mice (Fig 6A). In female mice, the expression of chemokines in the Wg was similarly increased in F-HFD compared with that in F-Chow mice. In contrast, their expression in OVX-HFD mice varied. In this context, expression of some chemokines, such as CCL3, 5, 2, and 22, were further increased, whereas that of other chemokines,

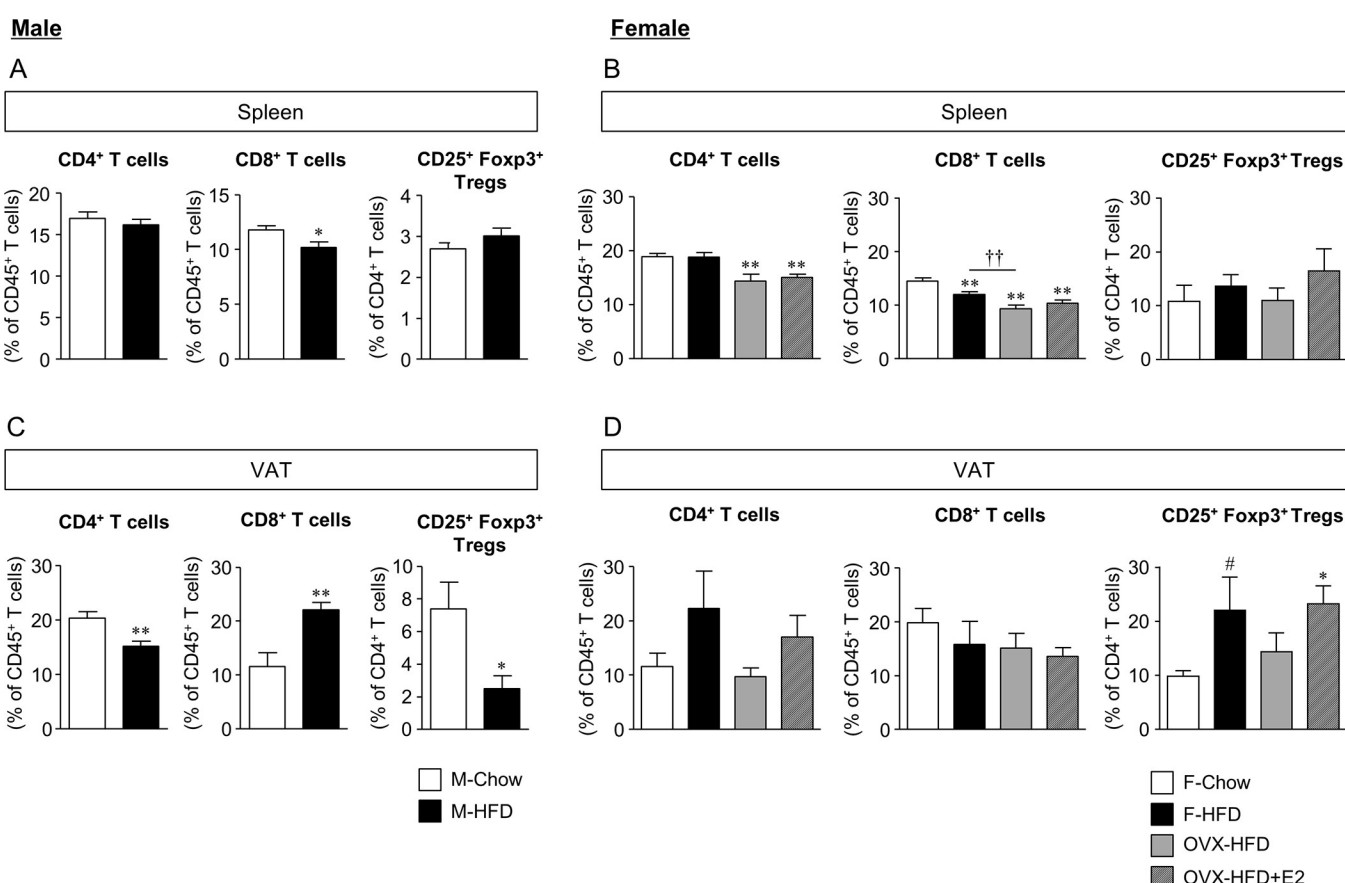

**Fig 3. Sex difference in the impact of HFD feeding on adipose tissue localization of CD4⁺ and CD8⁺ T cells and Treg.** The ratios of CD4⁺ and CD8⁺ T cells in CD45⁺ cells, and CD25⁺FOXP3⁺ Treg in CD4⁺ cells of spleen (A, B) and gonadal white adipose tissue (C, D) of male and female mice examined by flow cytometry are shown. Data are the mean ± S.E. (Spleen $n$ = 5–9, Wg $n$ = 4–6). *$P$<0.05 and **$P$<0.01, significantly different from control mice; #$P$<0.1 compared with control mice. The results of analysis with absolute cell number are shown in S2 Fig.

including CCL24, 20, and CXCL10, was not altered or even slightly reduced (Fig 6B). The chemokine expression in OVX-HFD+E2 mice also varied. In contrast, an almost consistent change was induced by HFD feeding in the expression of chemokine receptors in adipose CD4⁺CD25⁺ T cells (Fig 6A and 6B). In general, the expression decreased in both male and female mice fed HFD. Among them, characteristic expression change was observed in CCR4 of M-HFD mice, and in CCR6 and CXCR3 of female mice fed HFD.

Although adipose Treg decreased in the Wg of M-HFD mice, their number increased in F-HFD mice and OVX-HFD+E2 mice, and was not altered in OVX-HFD mice (Fig 3). We selected several genes that coordinately alter their expression according to the distribution of adipose Treg in female mice based on the heat map, and confirmed their expression by increasing the number of samples (Fig 6C and 6D). We also analyzed the expression of genes in male mice, and confirmed that *Ccl24* and *Cxcl10* expression increased and that of *Ccl20* slightly increased in the Wg, whereas that of *Ccr3*, *6* and *Cxcr3* slightly decreased in adipose CD4⁺CD25⁺ T cells of M-HFD mice (Fig 6C and 6D). In females, *Ccl24* expression was increased in the Wg of F-HFD and OVX-HF+E2 mice, whereas expression of its receptor *Ccr3* was not altered among adipose CD4⁺CD25⁺ T cells of each mouse group. Expression of *Ccl20* was not changed, and *Cxcl10* expression was instead increased in the Wg of OVX-HFD mice. In contrast, the expression of their corresponding receptors for *Ccr6* and *Cxcr3* in adipose

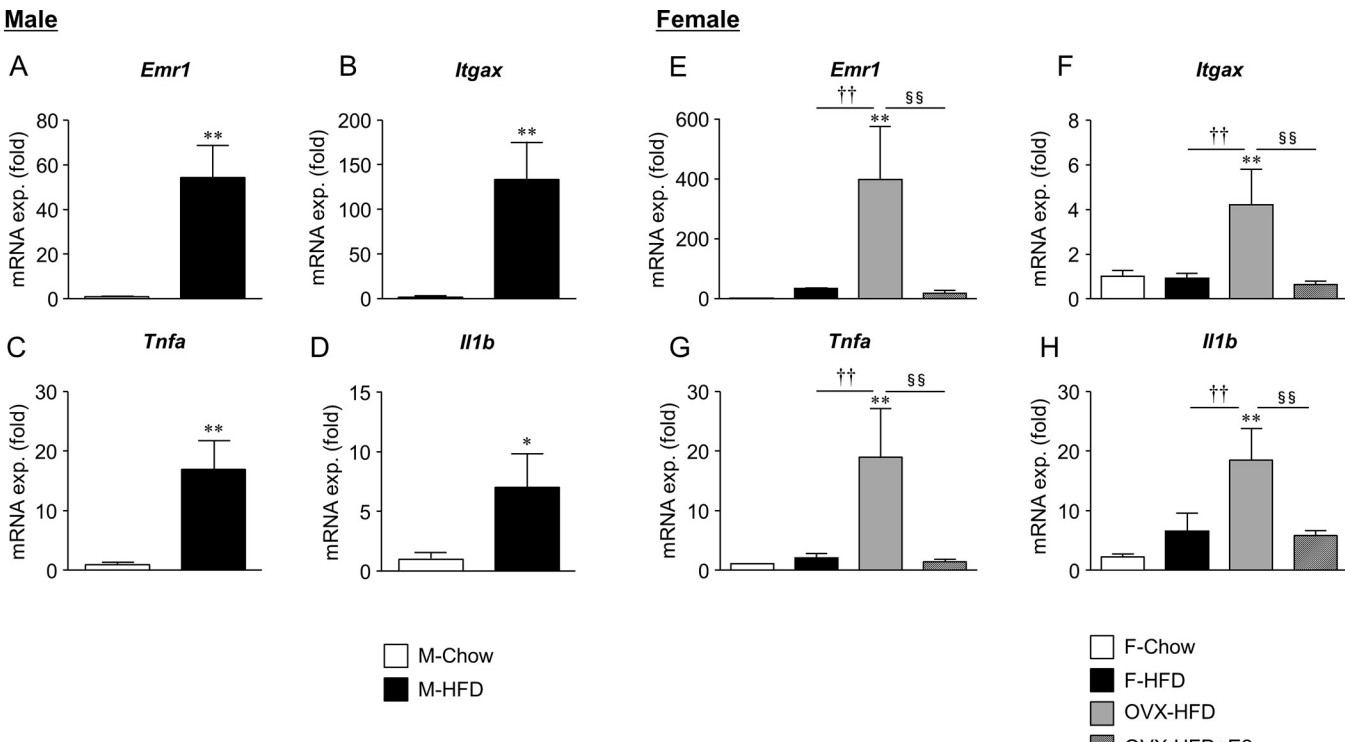

**Fig 4. Sex difference in the mRNA expression of inflammatory genes in Wg.** mRNA expression of *Emr1* (A, E), *Itgax* (B, F), *Tnfa* (C, G), and *Il1b* (D, F) in gonadal WAT of male and female mouse is shown. Data are the mean ± S.E. (*n* = 5–9). **$P < 0.01$, significantly different from control mice; ††$P < 0.01$, significantly different between F-HFD and OVX-HFD mice; §§$P < 0.01$, significantly different between OVX-HFD and OVX-HFD+E2 mice.

CD4+CD25+ T cells correlated with VAT-Treg accumulation. Their expression increased or slightly increased in both F-HFD and OVX-HFD+E2 mice, but not in OVX-HFD mice ([Fig 6E and 6F]). Obesity-associated alteration of inflammatory gene expression does not occur at the systemic levels because no notable changes in chemokine receptor expression were observed in the spleen among all groups of mice tested ([S3 Fig]).

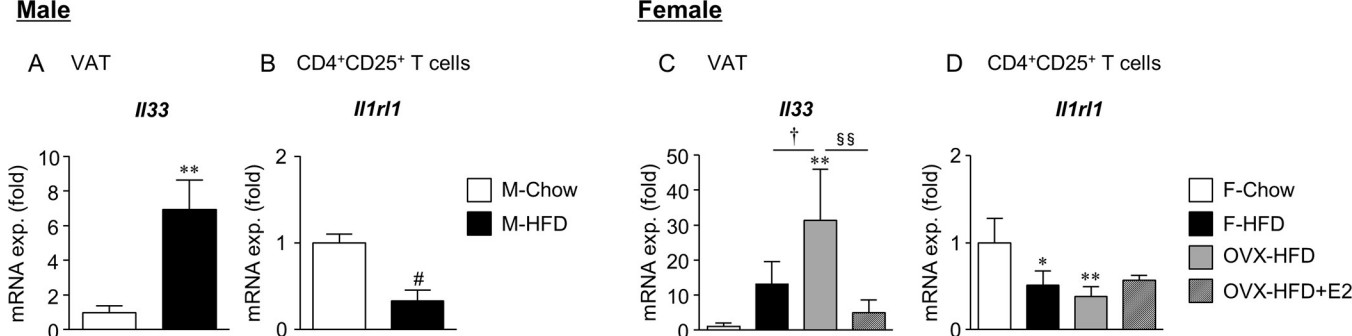

**Fig 5. Sex difference in the mRNA expression of IL33 in Wg and ST2 in adipose CD4+CD25+ T cells.** CD4+CD25+ T cells were isolated from the Wg of male and female mice by FACSAria cell sorter. mRNA expression of *Il33* in the Wg (A, C) and *Il1rl1* in CD4+CD25+ T cells (B, D) are shown. Data are the mean ± S.E. (*n* = 5–9). *$P < 0.05$ and **$P < 0.01$, significantly different from Chow mice; #$P < 0.1$, different from Chow mice; †$P < 0.05$, significantly different between F-HFD and OVX-HFD mice; §§$P < 0.01$, significantly different between OVX-HFD and OVX-HFD+E2 mice.

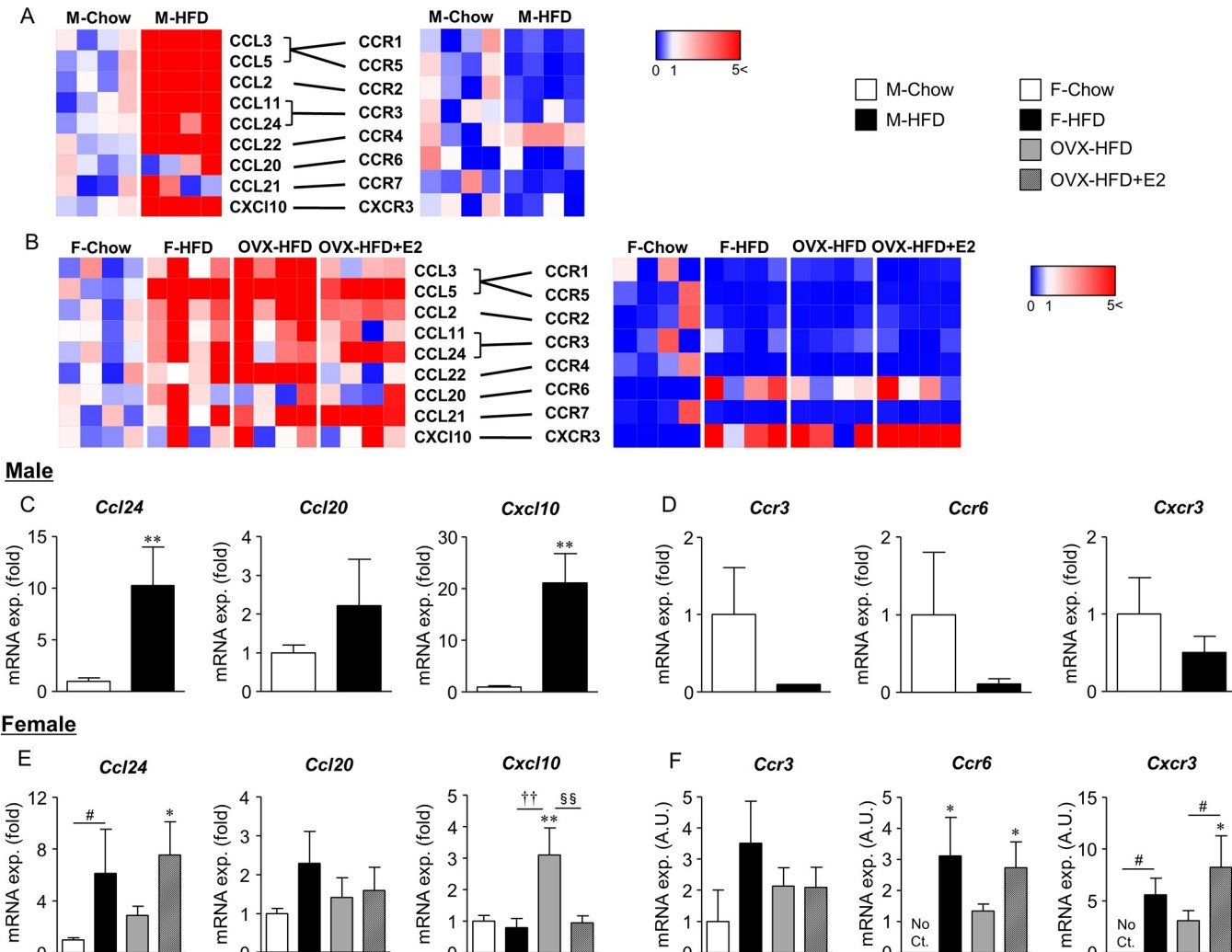

**Fig 6. Heat map of mRNA expression of chemokines in the Wg and their receptors in adipose CD4$^+$CD25$^+$ T cells.** CD4$^+$CD25$^+$ T cells were isolated from the Wg of male (A, C, D) and female (B, E, F) mice by FACSAria cell sorter. mRNA expression of chemokines in the Wg and their corresponding receptors in CD4$^+$CD25$^+$ T cells was analyzed by real-time PCR. Heat map analysis showed different gene expression pattern in males and females of each mouse group. Color from red to blue indicates high to low expression. mRNA expression of *Ccl24*, *Ccl20*, and *Cxcl10* in the Wg (C, E), and *Ccr3*, *Ccr6*, and *Cxcr3* in CD4$^+$CD25$^+$ T cells (D, F) is shown. Data are the mean ± S.E. ($n$ = 5–9; C-F). *$P<0.05$ and **$P<0.01$, significantly different from Chow mice; #$P<0.1$, among two groups, as indicated; ††$P<0.01$, significantly different between F-HFD and OVX-HFD mice; §§$P<0.01$, significantly different between OVX-HFD and OVX-HFD+E2 mice.

## Discussion

The decrease in estrogen in menopause is associated with increased risk of obesity and type 2 diabetes in women [22]. Physiological estrogens function in the maintenance of metabolic homeostasis by regulating energy homeostasis, insulin sensitivity, lipogenesis, and chronic inflammation in numerous tissues in females [16, 23–25]. Therefore, females are generally considered less susceptible to chronic inflammation associated with obesity, mainly due to the presence of estrogen. In addition, the obesity-associated decrease in VAT-Treg is also involved in chronic inflammation and impaired glucose metabolism in males [4]. However, the metabolic impact and effects of estrogen on the localization and function of VAT-Treg in females are unclear. In the current study, we demonstrated the opposite impact of obesity on the accumulation of VAT-Treg between male and female mice. Moreover, the increase in female was

inhibited in OVX-HFD mice, but was restored by supplementation with estrogen (Fig 3), suggesting an important role for estrogen in VAT-Treg accumulation in obesity. Furthermore, the expression of chemokine signaling molecules, including CCL24, CCR6, and CXCR3, was altered in accordance with female-specific fluctuations in VAT-Treg in obesity (Fig 6). Therefore, these chemokine signals are candidates mediating the accumulation of VAT-Treg.

The significance of IL33/ST2 signaling in the localization of VAT-Treg in male mice was previously demonstrated [20, 21, 26]. We also observed a reduction of VAT-Treg and decrease in ST2 expression in adipose CD4+CD25+ T cells despite the increase in IL33 in the Wg of M-HFD mice (Figs 3 and 5), as previously reported [20]. Of note, this decrease in ST2 was also observed in the adipose CD4+CD25+ T cells of F-HFD and OVX-HFD mice, suggesting the existence of a mechanism other than IL33/ST2 signaling in VAT-Treg localization in female mice. Recently, T cell receptors (TCR) in Treg that recognize VAT-specific antigens have been suggested as an important mechanism for VAT-Treg localization. In males, significant accumulation of VAT-Treg was reported in a transgenic mouse overexpressing the TCR gene of VAT-Treg clone, whereas no such findings were observed in the female mice [26]. This also suggests female-specific mechanisms for VAT-Treg accumulation.

Naturally occurring Treg differentiate in the thymus, and are distributed mainly to lymphoid tissues and throughout the body. Certain Treg migrate to peripheral tissues and play a homeostatic role in response to the tissue environment [27]. Tissue-resident Treg exhibit tissue-specific expression of chemokine receptors that mobilize them to target tissues by chemokine signals [28]. The expression of CCL24 in the Wg, and CCR6 and CXCR3 in adipose CD4+CD25+ T cell fluctuate in parallel with the number of VAT-Treg in female mice (Fig 6), suggesting the involvement of these signals in VAT-Treg recruitment. Indeed, the migration ability was decreased in Treg of CCR6 knockout mice *in vitro* [29]. Reduced Treg migration was also observed in the central nervous systems of the experimental autoimmune encephalomyelitis model Rag1 knockout mice reconstituted with bone marrow of CCR6 knockout mice [29]. Moreover, a decrease in FOXP3 expression was reported in the adipose tissue of male CXCR3 knockout mice fed HFD [30]. CXCR3 has been also demonstrated to be involved in the chemotaxis of Treg to pancreatic islets [13]. Regarding the effects of estrogen, it increases CCR6 expression in lymphoblasts *in vitro* [31]. As current limited knowledge regarding the molecular mechanism of estrogen and chemokine signaling for Treg mobilization should be further clarified in the future. We screened several chemokines and their receptors, and provided several candidate of chemokine signaling implicating VAT-Treg localization especially in females. It would be important to verify the expression of these receptors on Tregs by analysis with flow cytometry. Furthermore, the impact of chemokine receptor intervention identified in this study on VAT-Treg localization in female mice is needed to be clarified.

Group 2 innate lymphoid cells (ILC2) have recently been highlighted as immune cells controlled by sex hormones. Male are less susceptible to allergic airway inflammation, have a lower prevalence of asthma, and have lower ILC2 numbers in the lung and circulation compared with female mice and asthma patients. The negative impact of androgen and testosterone signaling has been suggested as a mechanism for the reduction in males [32, 33]. Since recent evidences suggest that ILC2 play crucial roles in adipose tissue homeostasis and browning [34], it would be interesting to investigate the sex difference of adipose ILC2 in various conditions such as obesity or cold exposure.

Estrogen replacement in mice is generally conducted by osmotic pump or sustained-release tablet [16, 23, 35], which is not physiological administration considering estrous cycles. The rodent estrus cycle repeats every 4 days, and a method of administering estrogen at 2 µg every four days has been reported as a physiological replacement for rats [15]. However, this dose was excessive as marked weight loss was observed. After careful induction in preliminary

experiments, we found 1.5 μg/mice to be ideal for estrogen replacement in C57BL/6 mice because it restored the estrus cycle based on vaginal smears, body fat gain, and disturbed energy and glucose metabolisms of OVX-HFD mice to those similar to HFD mice (Figs 1 and 2).

Several studies have attempted to elucidate the impact of obesity on the localization of VAT-Treg in humans. Contradictory results have been reported, possibly due to the difference in population, including age, menopause ratio, sex proportion, and methods for evaluating VAT-Treg utilizing flow cytometry or real-time PCR in each study [8, 9, 36–39]. The increase in VAT-Treg in obesity has been reported in studies where most subjects were women [36, 40]. In this context, the gender difference in VAT-Treg has also been described in obese mice [41], but the underlying mechanisms of sexual dimorphism or estrogen impact on VAT-Treg localization have not been clarified. The current study provides evidence that estrogen at a physiological level significantly affects VAT-Treg localization. Its effects may be one of the protective mechanisms to alleviate metabolic stress associated with obesity.

FOXP3 is an important transcription factor for the immuno-suppressive function of Treg; therefore, they are generally defined as $CD4^+CD25^+FOXP3^+$ T cells in mice [42]. As cell fixation with paraformaldehyde for intracellular staining of FOXP3 promotes nucleic acid fragmentation and is not suitable for the accurate analysis of mRNA expression [43], we isolated $CD4^+CD25^+$ T cells from the Wg and analyzed chemokine receptor expression (Fig 6). $CD4^+CD25^+$ T cells were considered Treg until the discovery of FOXP3 [44]. As recent studies demonstrated that some population of activated conventional T cells also express CD25 [45], experiments with Foxp3 reporter mice may be more ideal for the analysis of gene expression in tissue-resident Treg, being a limitation of the current study.

In summary, the current study demonstrated the sexual dimorphism in VAT-Treg accumulation in obesity. The increase in VAT-Treg in obesity may be induced by altered chemokine signals regulated by estrogen, which attenuates obesity-associated chronic inflammation and dysregulation of glucose metabolism in female mice.

## Supporting information

**S1 Fig. Gating strategy for $CD4^+CD25^+Foxp3^+$ Treg cells.** Representative plots of flow cytometry showing the gating strategy for identifying Tregs.
(TIF)

**S2 Fig. The impact of HFD feeding on absolute cell numbers of $CD4^+$ and $CD8^+$ T cells and Treg in the spleen and Wg (Related to Fig 3).** The absolute cell number ratios of $CD4^+$ and $CD8^+$ T cells and $CD4^+CD25^+FOXP3^+$ Treg in the spleen and gonadal white adipose tissue (Wg) of male and female mice examined by flow cytometry are shown. The results were obtained in the same experiments as in Fig 3. Data are the mean ± S.E. (Spleen $n$ = 5–9, Wg $n$ = 4–6). *$P<0.05$ and **$P<0.01$, significantly different from control mice; **$P<0.01$, significantly different from control mice; †$P<0.05$, significantly different between F-HFD and OVX-HFD mice; §§$P<0.01$, significantly different between OVX-HFD and OVX-HFD+E2 mice.
(TIF)

**S3 Fig. Heat map of mRNA expression of chemokine receptors in splenic $CD4^+CD25^+$ T cells.** $CD4^+CD25^+$ T cells were isolated from spleens of male (A, C) and female (B, D) mice by FACSAria cell sorter. mRNA expression of chemokine receptors in $CD4^+CD25^+$ T cells were analyzed by real-time PCR. Heat map analysis showing similar gene expression pattern in males and females of each mouse group. Color from red to blue indicates high to low

expression. mRNA expression of *Ccr3*, *Ccr6*, and *Cxcr3* in CD4$^+$CD25$^+$ T cells is shown. Data are the mean ± S.E. ($n$ = 4–9; C, D).
(TIF)

**S1 Data.**
(XLSX)

## Author Contributions

**Conceptualization:** Tsutomu Wada.

**Data curation:** Akari Ishikawa, Tsutomu Wada, Akira Okekawa, Yasuhiro Onogi, Eri Watanabe, Hiroshi Tsuneki, Shigeru Saito, Toshiyasu Sasaoka.

**Formal analysis:** Akari Ishikawa, Tsutomu Wada, Sanshiro Nishimura, Tetsuo Ito, Akira Okekawa, Azusa Sameshima, Tomoko Tanaka, Hiroshi Tsuneki.

**Funding acquisition:** Tsutomu Wada.

**Investigation:** Akari Ishikawa, Tsutomu Wada, Sanshiro Nishimura, Tetsuo Ito.

**Methodology:** Akari Ishikawa, Yasuhiro Onogi, Eri Watanabe, Azusa Sameshima, Tomoko Tanaka.

**Project administration:** Tsutomu Wada, Toshiyasu Sasaoka.

**Supervision:** Shigeru Saito.

**Validation:** Toshiyasu Sasaoka.

**Writing – original draft:** Akari Ishikawa, Tsutomu Wada, Toshiyasu Sasaoka.

**Writing – review & editing:** Tsutomu Wada, Toshiyasu Sasaoka.

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
