## [Decision Letter · Decision Letter 0]

17 Dec 2019

PONE-D-19-30423

Estrogen regulates sex-specific localization of regulatory T cells in adipose tissue of obese female mice

PLOS ONE

Dear Prof. Sasaoka,

Thank you for submitting your manuscript to PLOS ONE. After careful consideration, we feel that it has merit but does not fully meet PLOS ONE’s publication criteria as it currently stands. Therefore, we invite you to submit a revised version of the manuscript that addresses the points raised during the review process.

There were major issues identified by reviewer 1 that require significant attention, additional experiments, and re-write (especially within the methods section).

We would appreciate receiving your revised manuscript by Jan 31 2020 11:59PM. To enhance the reproducibility of your results, we recommend that if applicable you deposit your laboratory protocols in protocols.io, where a protocol can be assigned its own identifier (DOI) such that it can be cited independently in the future. For instructions see: http://journals.plos.org/plosone/s/submission-guidelines#loc-laboratory-protocols

We look forward to receiving your revised manuscript.

Kind regards,

Jonathan M Peterson, Ph.D.

Academic Editor

PLOS ONE

Journal Requirements:

Please ensure that your manuscript meets PLOS ONE's style requirements, including those for file naming. The PLOS ONE style templates can be found at http://www.plosone.org/attachments/PLOSOne_formatting_sample_main_body.pdf and http://www.plosone.org/attachments/PLOSOne_formatting_sample_title_authors_affiliations.pdfTo comply with PLOS ONE submission requirements, in your Methods section, please provide additional information regarding the experiments involving animals and ensure you have included details on (1) methods of sacrifice, (2) methods of anesthesia and/or analgesia, and (3) efforts to alleviate suffering.

This study was funded by the Japan Society for the Promotion of Science (JSPS KAKENHI Grant Number JP15K09410) to TW and a research grant from Mitsubishi Tanabe Pharma Corporation to TW.

We note that one of the authors received funding from a commercial source:Mitsubishi Tanabe Pharma Corporation

Additional Editor Comments (if provided):

There were major issues identified by reviewer 1 that require significant attention, additional experiments, and re-write (especially within the methods section).

Reviewers' comments:

Reviewer's Responses to Questions

**Comments to the Author**

1. Is the manuscript technically sound, and do the data support the conclusions?

Reviewer #1: No

Reviewer #2: Partly

2. Has the statistical analysis been performed appropriately and rigorously? 

Reviewer #1: N/A

Reviewer #2: Yes

3. Have the authors made all data underlying the findings in their manuscript fully available?

Reviewer #1: Yes

Reviewer #2: Yes

4. Is the manuscript presented in an intelligible fashion and written in standard English?

Reviewer #1: Yes

Reviewer #2: Yes

5. Review Comments to the Author

Reviewer #1: In this paper the authors provide some results suggesting that the recruitment of Treg cells in VAT upon HFD induced obesity is differentially regulated according the the sex, down regulated in male mice and up-regulated in female mice. This is an interesting observation, but the rest of the work suffers from flow in experimental designs and interpretation of the data.

Major concerns :

The results from Fig 1 to Fig 2 assessing the impact of ovarian hormones and E2 on HFD -induced metabolic phenotypes and on glucose tolerance are largely redundant to previously published works in this field.

The results in Fig 3 suggesting sex differences in the recruitment of Foxp3 Treg cells are interesting but very preliminary. The authors should show the gating strategy they used and provide a complete analysis of others immune cells associated with VAT, particularly ILC2. Sex bias have been shown regarding ILC2 numbers in various tissues (Cephus Cell report 2017; Laffont JEM 2017). They should also express the results as absolute cell numbers normalized to tissue weight.

The rationale of the experiment in Fig. 4 is unclear. Only TNFa is a pro-inflammatory cytokine gene. Emr1 and Itgax encode for F4/80 and CD11c, respectively. The expression of F4/80 and CD11c + cells must be assessed by FACS rather than RT-qPCR to be really informative..

Again, the rationale of Fig 5 is also unclear. Il1rl1 (ST2) IL33 receptor expression could be simply assessed by FACS on Tregs….

IL33 is an alarmin produced by epithelial cells undergoing cell death or necrosis, not by hematopoietic cells. There is no rationale to test IL33 mRNA expression in Treg cells….

In Fig 6 the data must be expressed as relative expression normalized to house keeping genes rather than fold-change which does not mean anything. Again, the expression of chemokine receptor (CCR3, CCR6, CXCR3) could be easily tested by FACS on Treg cells.

Reviewer #2: In the article the authors wanted to address the difference in obesity in male and female populations and the chronic inflammatory response related to obesity. The article also addresses how estrogen protect against the chronic inflammation in females through Treg cellular response. Overall the article was well written and the conclusions were well supported. However, there were a couple of significant but minor issues, specifically within the methods section, that need to be addressed prior to publication.

· Investigators have in methods section “as previously described” on lines 103, 110, 114, and 153. When reviewing the cited articles on line 103, #16 did not have any methods on MRI. Citation #17 and it did have MRI in its methods but it also said “as previously described” with a citation. I will assume the other citations are the same. It would be easier for the readers if the protocol is described and that the references indicated actually describe the methods being used.

· Similarly the details for the glucose tolerance test and insulin tolerance tests were incomplete. For example how long were the animals fasted prior to start of experiment?

· When reporting centrifugation within the methods section the authors must either indicated the Relative centrifugal force, or RCF, not RPMs, as without the rotor model number the RPM value cannot be repeated.

6. PLOS authors have the option to publish the peer review history of their article (what does this mean?). If published, this will include your full peer review and any attached files.

Reviewer #1: No

Reviewer #2: Yes: Kristy L. Thomas

---

## [Author Response · Author response to Decision Letter 0]

30 Jan 2020

Answers to Academic Editor

 The Editor indicated there were major issues identified by the Reviewer 1 that required significant attention, additional experiments, and re-write manuscript. We have carefully considered all of the comments of the Reviewers. We added gating strategy of Tregs (Fig. 1S), flow cytometry data of CD4+ and CD8+ T cells (Fig. 3), absolute cell counts for flow cytometry data (Fig. 2S), and measured Il1b expressions in the Wg of each mouse as another inflammatory cytokine (Fig. 4) in the revised manuscript. 

 The Reviewer 1 has requested several flow cytometry experiments. We understand analysis with flow cytometry provides more informative and straightforward results compared with mRNA analysis of sorted cells by flow cytometry. Ideally, all experiments should be performed by flow cytometry. However, we could not conduct additional flow cytometry experiments. There are several reasons why we could not confirm our results with flow cytometry, and we answered each of them in the Response to Reviewer 1. We would like to explain an overview here. The main reason is limited number of stromal vascular fraction (SVF) cells available in adipose tissues. The number of SVF cells in adipose tissue is low, especially in lean animals. Therefore, more than one mouse per one sample is needed for each analysis (macrophages, expression of ST2 or one chemokine receptor on Tregs). Our FACSArea II system (BD Bioscience) can measure only 6 colors simultaneously to analyze adipose tissue SVF cells with appropriate compensation. We have already used 5 colors (7AAD, CD45, CD4, CD25, and Foxp3) for Treg separation. In addition, we need strict consideration of experimental ethics regarding the number of mice required for the experiments. Furthermore, it would take 5 month to prepare mice for flow cytometry analysis, because mice were fed high-fat diet for 16 weeks after sham or ovariectomy. 

 Current study has shown the sex difference on VAT-Treg accumulation in obese condition. We screened mRNA expression of several chemokine receptors, and found synchronous changes in CCR3, CCR6, and CXCR3 expression on Tregs with VAT-Treg localization. Although we did not confirm their expressions by flow cytometry, we believe that the current findings are novel and meaningful for further understanding the localization of VAT-Treg. 

 We would like to thank the Editor and the Reviewers for highly evaluating our manuscript. Our detailed response to each of the Reviewer’s criticism is described bellows. We really hope that our revised manuscript is now acceptable for publication in PLOS ONE.

Answers to Reviewer 1

We thank the Reviewer for evaluating our manuscript and his/her thoughtful suggestions for improving our manuscript.

1. The results from Fig 1 to Fig 2 assessing the impact of ovarian hormones and E2 on HFD -induced metabolic phenotypes and on glucose tolerance are largely redundant to previously published works in this field.

 The Reviewer suggested that impacts of HFD feeding, results in Figs. 1 and 2 are redundant, since ovariectomy and estradiol (E2) treatment on body weight gain and glucose metabolism are already reported. Although the Reviewer’s indication is taken, we carefully re-consider that these results are still important basic data for properly understanding current study, because body fat accumulation, adipose Treg accumulation and glucose metabolism are closely related to each other. In addition, we did provide a novel method for E2 supplementation in mice, which was modified from a previous reported study on rats [Reference 15, Horm Behav. 2002 doi: 10.1006/hbeh.2002.1835.]. We administered 1.5 µg of E2 every 4 days to imitate the estrus cycle of mice. Most of metabolic parameters including body and fat weights, energy metabolism, glucose levels in glucose and insulin tolerance test were comparable between female sham-operated HFD-fed mice (F-HFD) and E2-supplemented ovariectomized mice fed HFD (OVX-HFD+E2), indicating that the method of E2 supplementation appears to be adequate in the experiments. Although continuous E2 supplementation methods such as sustained releasing tablets or osmotic pump are commonly used for mice experiments, we believe the current new method is considered simple, cost-effective and physiological approach. Therefore, we understand that demonstration of results in Figs. 1 and 2 is important for verifying the data with the current supplementation method.

2. The results in Fig 3 suggesting sex differences in the recruitment of Foxp3 Treg cells are interesting but very preliminary. The authors should show the gating strategy they used and provide a complete analysis of others immune cells associated with VAT, particularly ILC2. Sex bias have been shown regarding ILC2 numbers in various tissues (Cephus Cell report 2017; Laffont JEM 2017). They should also express the results as absolute cell numbers normalized to tissue weight.

 The Reviewer indicated that sex differences in the recruitment of Foxp3 adipose Treg cells in Fig. 3 are interesting but preliminary. VAT-Treg plays significant role in the attenuation of obesity-associated chronic inflammation, thereby contributing to the maintenance of systemic glucose metabolism in male animals [Reference 7, Nat Med. 2009, doi: 10.1038/nm.2002.; Reference 8, Nat Med. 2009, doi: 10.1038/nm.2001.]. Therefore, we think that the observation is interesting as the Reviewer indicated. In the following experiments, we attempted to investigate the possible underlying mechanisms of VAT-Treg accumulation in female mice by screening their chemokine receptor expressions, since distribution of resident Treg has been shown to be regulated by chemokine signals in various tissues [Reference 27, Blood. 2006, doi: 10.1182/blood-2006-01-0177.]. We really hope that the Reviewer understands our research focus and strategy.

 In response to the Reviewer’s request, we added gating strategy of Treg analysis, which was added as the S1 Fig. and described in the method section of the revised manuscript (page 10, line 145).

 The Reviewer suggested to provide complete analysis of other immune cells associated with VAT, particularly ILC2, since their sex bias has been shown recently. Therefore, we demonstrated CD4+ and CD8+ T cell data in the spleen and gonadal white adipose tissue (Wg) in addition to Tregs in Fig. 3 and results section of the revised manuscript (page 15, line 211 to page 16, line 219), although we could not provide complete analysis of immune cells including ILC2. We understand that the Reviewer’s suggestion about ILC2 analysis is quite interesting, because recent studies have indicated that adipose ILC2 plays significant roles in tissue homeostasis and browning. However, preparation of new separate sets of each mouse group is required for the complete analysis of immune cells including ILC2. It will take about more than 5 month for the preparation of mice, because mice were loaded HFD for 16 weeks after ovariectomy. Since we really understand that the adipose ILC2 analysis is another important research topic in the future, we proposed ILC2 experiment as a crucial and needed topic in the discussion section of the revised manuscript (page 25, lines 362 to 369), as follows: Group 2 innate lymphoid cells (ILC2) have recently been highlighted as immune cells controlled by sex hormones. Male are less susceptible to allergic airway inflammation, have a lower prevalence of asthma, and have lower ILC2 numbers in the lung and circulation compared with female mice and asthma patients. The negative impact of androgen and testosterone signaling has been suggested as a mechanism for the reduction in males [32, 33]. Since recent evidences suggest that ILC2 play crucial roles in adipose tissue homeostasis and browning [34], it would be interesting to investigate the sex difference of adipose ILC2 in various conditions such as obese or cold stimulation. 

 The Reviewer suggested to explain the results as absolute cell numbers normalized to tissue weight in Fig. 3. The presentation of flow cytometry data with cell frequency or absolute cell numbers is sometimes controversial. We considered cell frequency data to be useful because of the low variability and elimination of dead cells. In this context, we presented Fig. 3 as the data with cell frequency. In contrast, data with absolute cell number is also important especially for the analysis with rarely localized cell types. Indeed, frequencies of lung ILC2 cells are rare (about less than 1% in both papers: Cephus Cell report 2017 and Laffont JEM 2017). We understand that presentation of data with both frequency and absolute cell numbers is ideal, as the Reviewer suggested. In this context, the absolute numbers of adipose Treg did not change significantly in female F-HF and OVX-HFD+E2 mice and even in any group of male mice, possibly due to remarkable increase of CD4+ T cells by HFD feeding in our experimental condition. Since adipose Treg is usually evaluated by frequency data [Reference 7, Nat Med. 2009, doi: 10.1038/nm.2002.; Reference 8, Nat Med. 2009, doi: 10.1038/nm.2001.] and the interpretation of absolute cell numbers data is difficult, we added these data in the S2 Fig. of the revised manuscript.

3. The rationale of the experiment in Fig. 4 is unclear. Only TNFa is a pro-inflammatory cytokine gene. Emr1 and Itgax encode for F4/80 and CD11c, respectively. The expression of F4/80 and CD11c + cells must be assessed by FACS rather than RT-qPCR to be really informative.

 Obesity-associated infiltration of proinflammatory macrophage is a well-known feature of chronic inflammation in the visceral adipose tissue that produces inflammatory cytokines including TNFα. The Reviewer suggested that only Tnfa is a pro-inflammatory cytokine gene shown in Fig. 4 of the original manuscript. Since we agree with the Reviewer’s indication, we analyzed the expressions of Il1b as another important cytokine known to be implicated in the development of insulin resistance in the obese adipose tissue. The expression showed similar change compared to Tnfa in the mice group. These new results are added in the Fig. 4 and stated in the result section of the revised manuscript (page 17, lines 231 to 232), as follow: the expression of macrophage markers Emr1and Itgax,and proinflammatory cytokines Tnfa and Il1b were significantly increased (Fig. 4A-D).

 The Reviewer indicated that Emr1 and Itgax encoding F4/80 and CD11c are representative markers for macrophages and proinflammatory M1-macrophages, respectively. The Reviewer suggested to assess these cell numbers by flow cytometry. We understand that analysis with flow cytometry is ideal for evaluation of macrophage infiltration. However, the number of SVF cells in adipose tissue is low especially in lean animals. More than one mouse per one sample is needed only for the macrophage analysis. In addition, it takes more than 5 months to prepare the new separate set of each mouse group for the assay. We really hope that the Reviewer understands the practical difficulties of analyzing adipose tissue with flow cytometry. In contrast, mRNA expressions of Emr1 and Itgax are usually analyzed as adequate indicators of macrophage infiltration in adipose tissue (e.g. Takei R, PLOS One 2019, doi: 10.1371/journal.pone.0223302.; Li J, Nat Commun 2019, doi: 10.1038/s41467-019-10348-0.; Kawano Y, Cell Metab 2016, doi: 10.1016/j.cmet.2016.07.009.). Therefore, we measured these mRNA expressions as indicators of obesity-associated chronic inflammations in these animals. Again, we really hope that the Reviewer understands our careful decision on the experiments.

4. Again, the rationale of Fig 5 is also unclear. Il1rl1 (ST2) IL33 receptor expression could be simply assessed by FACS on Tregs….

IL33 is an alarmin produced by epithelial cells undergoing cell death or necrosis, not by hematopoietic cells. There is no rationale to test IL33 mRNA expression in Treg cells….

 The significance of IL33/ST2 signaling in the localization of VAT-Treg in male mice has been previously demonstrated [References 20, 21, 26 of the revised manuscript]. Therefore, we investigated whether the expression of Il1lr1 is associated with VAT-Treg localization in female mice, and found that they were not correlated as described in the discussion section. The Reviewer again suggested that investigation of ST2 (Il1rl1) expression in Treg could be assessed by flow cytometry rather than mRNA analysis in sorted samples. We really understand the importance of ideal analysis with flow cytometry for ST2 expression in Treg. Again, we would like to mention that the number of SVF cells in adipose tissue is limited. Preparation of a new separate set of samples from each mouse group is practically very difficult for the assay. On the other hand, we believe that the measurement of mRNA expression is an adequate approach for analyzing limited samples with appropriate biological implications.

 We measured IL33 levels in the visceral adipose tissue, since it is an alarmin, as the Reviewer indicated. We labeled “VAT” and “CD4+CD25+ T cells” above panels to indicate analyzed samples in the revised Fig. 5.

5. In Fig 6 the data must be expressed as relative expression normalized to house keeping genes rather than fold-change which does not mean anything. Again, the expression of chemokine receptor (CCR3, CCR6, CXCR3) could be easily tested by FACS on Treg cells.

 All of expression data including Fig. 6 are expressed as relative expression normalized to the 18S ribosomal RNA, as described in the material and method section. 

 Current study has shown the sex difference on VAT-Treg accumulation in obese condition. We screened mRNA expression of several chemokine receptors, and found synchronous changes in CCR3, CCR6, and CXCR3 expression on Tregs with VAT-Treg localization. Although we did not confirm their expressions by flow cytometry analysis, we believe that the current findings are novel and valuable for further understanding the mechanism of VAT-Treg localization. At the same time, we understand the limitation that current study shows a possible involvement of these chemokine signaling in the clarification of VAT-Treg localization mechanisms in female. The limitation of the current study and requirement of the suggested future research has been described in the discussion section of the manuscript. Furthermore, since the Reviewer’s indication that chemokine receptor expressions should be confirmed by flow cytometry analysis is well taken, we included this point in the discussion section of the revised manuscript (page 25, lines 355 to 361), as shown below: As current limited knowledge regarding the molecular mechanism of estrogen and chemokine signaling for Treg mobilization should be further clarified in the future. We screened several chemokines and their receptors, and provided several candidate of chemokine signaling implicating VAT-Treg localization especially in females. It would be important to verify the expression of these receptors on Tregs by analysis with flow cytometry. Furthermore, the impact of chemokine receptor intervention identified in this study on VAT-Treg localization in female mice is needed to be clarified.

Answers to Reviewer 2

We thank the Reviewer for highly evaluating our manuscript and his/her thoughtful suggestions for improvement of our manuscript.

1. The Reviewer suggested to revise references in the method section to adequately repeat the experiments for readers. According to the Reviewer’s important suggestion, we have carefully checked all of references regarding the adequate explanation of the experimental protocol in the method section. As a result, we deleted references 16 and 19 in original manuscript, and adequately corrected and added one paper (reference 17) in the revised manuscript.

2. The Reviewer asked to describe fasting time of GTT and ITT experiments. We agree with the Reviewer’s indication, since fasting time is important for understanding of GTT and ITT data. In response to the Reviewer’s comment, we stated the fasting time in the method section of the revised manuscript (page 8, line 115 to page 9, line 117). 

3. The Reviewer advised to describe the relative centrifugal force when reporting centrifugation in the method section. According to the Reviewer’s comment, we changed the description about centrifugation from RPM to RCF in the revised manuscript (Page 9, lines 122 and 128).

---

## [Decision Letter · Decision Letter 1]

11 Mar 2020

Estrogen regulates sex-specific localization of regulatory T cells in adipose tissue of obese female mice

PONE-D-19-30423R1

Dear Dr. Sasaoka,

We are pleased to inform you that your manuscript has been judged scientifically suitable for publication and will be formally accepted for publication once it complies with all outstanding technical requirements.

With kind regards,

Jonathan M Peterson, Ph.D.

Academic Editor

PLOS ONE

Additional Editor Comments (optional):

all comments from previous reviews have been answered. While additional experiments would enhance the findings of the manuscript, they are not required.

Reviewers' comments:

Reviewer's Responses to Questions

**Comments to the Author**

1. If the authors have adequately addressed your comments raised in a previous round of review and you feel that this manuscript is now acceptable for publication, you may indicate that here to bypass the “Comments to the Author” section, enter your conflict of interest statement in the “Confidential to Editor” section, and submit your "Accept" recommendation.

Reviewer #2: All comments have been addressed

Reviewer #3: All comments have been addressed

2. Is the manuscript technically sound, and do the data support the conclusions?

Reviewer #2: Yes

Reviewer #3: Partly

3. Has the statistical analysis been performed appropriately and rigorously? 

Reviewer #2: Yes

Reviewer #3: Yes

4. Have the authors made all data underlying the findings in their manuscript fully available?

Reviewer #2: Yes

Reviewer #3: Yes

5. Is the manuscript presented in an intelligible fashion and written in standard English?

Reviewer #2: Yes

Reviewer #3: Yes

6. Review Comments to the Author

Reviewer #2: Authors have sufficiently addressed the minor revisions addressed in the previous review. There are no additional revisions that have appeared in this newest draft.

Reviewer #3: Q1: Circulating adipokines play a critical role in systemic inflammation and insulin resistance. The authors' work and responses to reviewers have well done. I'm wondering if it is possible to look at serum adipokine profile?

Q2: The conclusion would be solider if in vitro experiments could be conducted to investigate the mechanism by which estrogen acts to IL33/ST2 in the context of nutrient excess.

7. PLOS authors have the option to publish the peer review history of their article (what does this mean?). If published, this will include your full peer review and any attached files.

Reviewer #2: Yes: Kristy L. Thomas

Reviewer #3: No

---

## [Editor Report · Acceptance letter]

18 Mar 2020

PONE-D-19-30423R1 

Estrogen regulates sex-specific localization of regulatory T cells in adipose tissue of obese female mice 

Dear Dr. Sasaoka:

I am pleased to inform you that your manuscript has been deemed suitable for publication in PLOS ONE. Congratulations! Your manuscript is now with our production department. 

With kind regards,

on behalf of

Dr Jonathan M Peterson 

Academic Editor

PLOS ONE